

# Classification of ovarian cancer associated with BRCA1 mutations, immune checkpoints, and tumor microenvironment based on immunogenomic profiling

Yousheng Wei[1,*], Tingyu Ou[1,*], Yan Lu[1], Guangteng Wu[1], Ying Long[1], Xinbin Pan[2] and Desheng Yao[1]

[1] Department of Gynecologic Oncology, Guangxi Medical University Cancer Hospital, Nanning, Guangxi, China

[2] Department of Radiation Oncology, Guangxi Medical University Cancer Hospital, Nanning, Guangxi, China

[*] These authors contributed equally to this work.

Corresponding author
Desheng Yao,
yaodesheng@gxmu.edu.cn

## ABSTRACT

**Background**. Ovarian cancer is a highly fatal gynecological malignancy and new, more effective treatments are needed. Immunotherapy is gaining attention from researchers worldwide, although it has not proven to be consistently effective in the treatment of ovarian cancer. We studied the immune landscape of ovarian cancer patients to improve the efficacy of immunotherapy as a treatment option.

**Methods**. We obtained expression profiles, somatic mutation data, and clinical information from The Cancer Genome Atlas. Ovarian cancer was classified based on 29 immune-associated gene sets, which represented different immune cell types, functions, and pathways. Single-sample gene set enrichment (ssGSEA) was used to quantify the activity or enrichment levels of the gene sets in ovarian cancer, and the unsupervised machine learning method was used sort the classifications. Our classifications were validated using Gene Expression Omnibus datasets.

**Results**. We divided ovarian cancer into three subtypes according to the ssGSEA score: subtype 1 (low immunity), subtype 2 (median immunity), and subtype 3 (high immunity). Most tumor-infiltrating immune cells and immune checkpoint molecules were upgraded in subtype 3 compared with those in the other subtypes. The tumor mutation burden (TMB) was not significantly different among the three subtypes. However, patients with BRCA1 mutations were consistently detected in subtype 3. Furthermore, most immune signature pathways were hyperactivated in subtype 3, including T and B cell receptor signaling pathways, PD-L1 expression and PD-1 checkpoint pathway the NF-$\kappa$B signaling pathway, Th17 cell differentiation and interleukin-17 signaling pathways, and the TNF signaling pathway.

**Conclusion**. Ovarian cancer subtypes that are based on immune biosignatures may contribute to the development of novel therapeutic treatment strategies for ovarian cancer.

## INTRODUCTION

Ovarian cancer has the highest fatality rate among gynecological malignancies. It is estimated that 21,750 new cases of ovarian cancer will be diagnosed in 2020, resulting in 13,940 ovarian cancer fatalities (*Siegel, Miller & Jemal, 2020*). Most ovarian cancer cases are diagnosed at an advanced stage due to a lack of overt symptoms, and the 5-year relative survival rate is only about 40% (*Bray et al., 2018*; *Lheureux et al., 2019*; *Torre et al., 2018*). This rate has only improved slightly over the past few decades thanks to advancements in research (*Ghisoni et al., 2019*; *Holmes, 2015*). The standard treatments for ovarian cancer include surgery and platinum-based chemotherapy. Complete remission can occur for most patients following their initial treatment but there is still a high rate of reoccurrence (*Odunsi, 2017*). Therefore, novel therapeutic approaches are needed to improve the quality of life and survival of these patients.

Cancer immunotherapy is a promising treatment for many types of solid tumors (*Bellmunt et al., 2017*; *Reck et al., 2016*). It eliminates cancer primarily by acting on the immune system or the tumor microenvironment but not on tumor cells directly. Cancer cells affect antigen presentation, disrupt the regulatory cascades of T cells, mobilize immune-suppressing cells, and produce active cytokines with immune repressive effects, thereby weakening the immune system, modifying immune regulation, and benefiting tumor cells (*Antonia, Larkin & Ascierto, 2014*; *Odunsi, 2017*). The immune checkpoint inhibitor (ICI)-based antibody has improved survival for patients with different types of cancer, including malignant melanoma, lung cancer, and bladder cancer. This antibody is directed at cytotoxic T lymphocyte-associated antigen-4 (CTLA-4), programmed cell death 1 (PD-1), and programmed cell death 1 ligand 1 (PD-L1) receptors; it initiates immune cell function, and normalizes the tumor microenvironment (*Bellmunt et al., 2017*; *Borghaei et al., 2015*; *Robert et al., 2015*). The response rate of ovarian cancer to ICIs is discouraging, with an objective response rate (ORR) of <15% (*Hamanishi et al., 2015*; *Matulonis et al., 2019*). In the phase II KEYNOTE-100 study of 376 patients with advanced recurring ovarian cancer, it was found that pembrolizumab monotherapy was linked to an ORR of 8.0% (95% CI [5.4–11.2]), and a higher PD-L1 expression level was also linked to a better response (*Matulonis et al., 2019*). Single agent ICIs have exhibited only modest improvements in this type of malignancy. In fact, genomic features, such as PD-L1 expression, tumor mutation burden (TMB), neoantigen load, and defects in DNA damage repair, have been associated with tumor immunotherapeutic responsiveness in ovarian cancer (*Ghisoni et al., 2019*; *Odunsi, 2017*; *Tian et al., 2020*).

We classified ovarian cancer based on 29 immune signatures, representing different immune cell kinds, functions, and pathways. Single-sample gene set enrichment (ssGSEA) was used to quantify the activity or enrichment levels of the gene sets in cancer, and ovarian cancer was classified into three subtypes: subtype 1 (low immunity), subtype 2 (median immunity), and subtype 3 (high immunity). We compared the tumor microenvironment, immune cells, immune checkpoint molecules, TMB, BRCA1/2 mutation, prognosis, gene ontology, and pathways. Our findings may assist in the selection of ovarian cancer patients who would benefit from immunotherapy.

## MATERIALS AND METHODS

### Data

Gene expression profiles were mined from The Cancer Genome Atlas (TCGA) dataset (https://tcga-data.nci.nih.gov/tcga/) and consisted of normalized gene expression patterns for 379 ovarian cancer samples that were mapped using fragments per kilobase of transcript per million fragments. Clinical data for age, survival, stage, and tumor grade were also mined from TCGA. The somatic mutation data were obtained from single nucleotide polymorphism (SNP) data in the TCGA repository using Mutect. The expression data of the validation dataset was obtained from the Gene Expression Omnibus (GEO) repository (GSE51088), and contained 172 ovarian cancer samples. All computational and statistical analyses were performed using R software (version 3.6.1, http://www.R-project.org).

### ssGSEA and clustering

We obtained 29 immune-associated gene sets, totaling 707 genes, representing different immune cell types, functions, and pathways (File S1) (*He et al., 2018*; *Yue, Ma & Zhou, 2019*). We used single sample gene set enrichment analysis (ssGSEA), using the R package GSVA (version 1.34.0), to calculate the enrichment scores of the 29 immune biosignatures for each sample in the tumor microenvironment (*Barbie et al., 2009*; *Hänzelmann, Castelo & Guinney, 2013*). ssGSEA calculated the gene signature overexpression scores by comparing the gene levels in the signature with those in all the other genes in the transcriptome. An unsupervised machine learning method was used to perform hierarchical clustering of ovarian cancer into three clusters. The clusters were further divided into three subtypes according to the immune scores: subtype 1, subtype 2, and subtype 3.

### ESTIMATE and CIBERSORT

Estimation of STromal and Immune cells in MAlignant Tumor tissues using Expression data (ESTIMATE) (*Yoshihara et al., 2013*) is an approach that uses gene expression biosignatures to deduce the proportion of stromal and immune cells in tumor samples, which form the major non-tumor constituents of tumor samples. By performing ssGSEA, stromal and immune scores are calculated to estimate the levels of invading stromal and immune cells. This forms the rationale for the ESTIMATE score to deduce tumor purity in the tumor tissue. Tumor purity was calculated based on the gene expression data from the immune and stromal scores using ESTIMATE in R package. Cell-type Identification by Estimating Relative Subsets of RNA Transcripts (CIBERSORT) (*Newman et al., 2015*) is a biological tool that uses the deconvolution strategy to compute the fractions of the 22 human immune cell types. We selected 1000 permutations and data with $P < 0.05$ as the maxim for the successful deconvolution of a sample. The Kruskal-Wallis test was used to compare the proportions of immune cell types among ovarian cancer subtypes.

### Calculation of TMB scores

TMB is the overall enumeration of mutations per million bases in tumor tissue. It details the mutation density of tumor genes, i.e., the enumeration of mutations in the tumor genome, including the total number of genetic coding errors, base substitutions, and gene insertions
or deletions. We computed the mutation frequency with the number of variants/the length of exons (38 million) for each sample using Perl (v5.30.1, https://www.perl.org/).

## Survival analyses

We obtained the follow-up information of patients from their clinical data and calculated the significance of survival time using the log-rank test and their differences using a threshold of $P < 0.05$. The median was used as the cut-off value to divide the samples into high- or low scores to obtain the relationship between the related immune gene sets score and prognosis. We plotted the Kaplan–Meier curves to indicate the differences in the survival periods.

## Gene-set enrichment analysis

Gene-set enrichment analysis of TCGA datasets was conducted in the GSEA (R package) (*Subramanian et al., 2005*). The Kyoto Encyclopedia of Genes and Genomes (KEGG) and Gene Ontology (GO) analyses were used to assess the functional role of the differentially expressed genes between subtype 1 and subtype 3. Differential gene set enrichment was inspected by the limma R package. $P < 0.05$ was used as the cut-off value.

# RESULTS

## Immunogenomic profiling identifies three ovarian cancer subtypes

We obtained enrichment scores for each sample of the 29 immune-associated gene sets in the tumor microenvironment using ssGSEA. Ovarian cancer cases were then hierarchically clustered into three classes according to their immune scores. These classes were clearly defined and classified as subtype 1, subtype 2, and subtype 3, which represented low immunity, median immunity, and high immunity, respectively (Fig. 1A). We determined the immune, stromal, ESTIMATE scores, and tumor purity using the ESTIMATE algorithm. Our results showed that the immune and stromal scores were significantly different ($P < 0.001$) and were highest in subtype 3 and lowest in subtype 1; tumor purity was the highest in subtype 1 and lowest in subtype 3 (Figs. 1B–1D). These results indicated that immune and stromal cells are most prevalent in subtype 3, while tumor cells are most prevalent in subtype 1.

We found that most of the human leukocyte antigen (HLA) gene expression levels were lowest in subtype 1 and highest in subtype 3 ($P < 0.001$) (Fig. S1A). The expression levels of many immune cell subgroup biomarker genes, such as FOXP3 [regulatory T cell (Treg)], CD45RO (memory T cell), CD8A (cytotoxic T cell), CD20 (B cell), CD1A [immature dendritic cell (iDC)], CXCR5 (Tfh cell), and IL3RA [plasmacytoid dendritic cell (pDC)] were remarkably higher in subtype 3 and markedly lower in subtype 1 (Fig. S1B).

## Three subtypes show differential expression of immune checkpoint genes

We analyzed the expression levels of the checkpoint receptors in the three ovarian cancer subtypes responsible for decreasing T cell bioactivity, including PDCD1 (PD1), CTLA4, LAG-3, and TIM-3. We then analyzed the PDCD1 ligand CD274 (PD-L1), PDCD1LG2 (PD-L2), CTLA4 ligand CD86, and CD80. We found that the expression levels of these 8

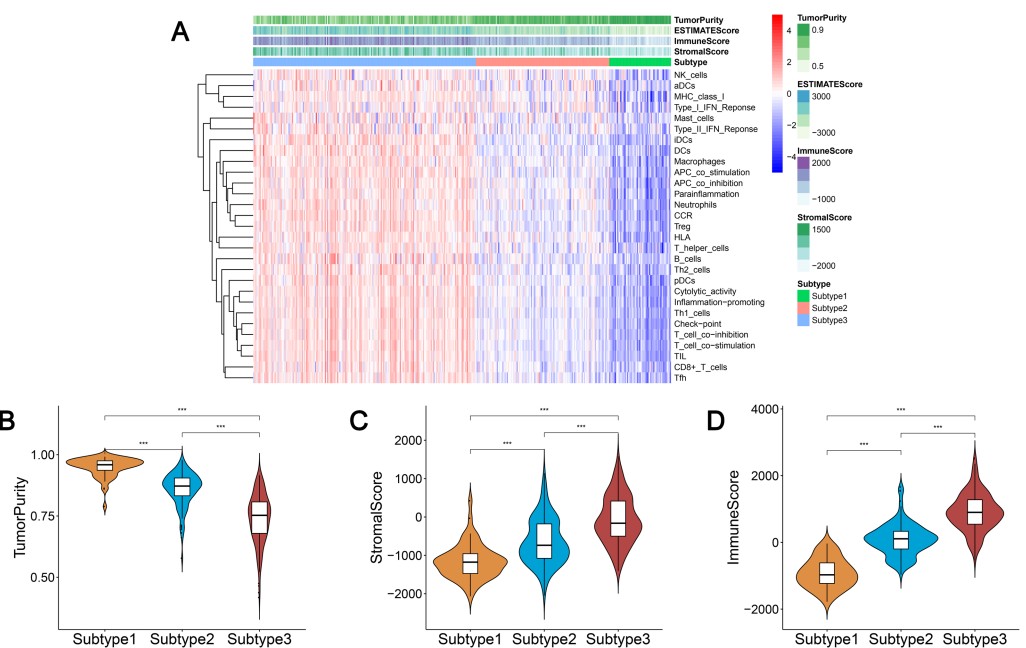

**Figure 1** **Immunogenomic profiling identifies three ovarian cancer subtypes.** (A) Ovarian cancer was hierarchically clustered into three clusters in The Cancer Genome Atlas dataset. In the heat map of gene expression, red represents high expression and blue represents low expression. Tumor purity, ESTIMATE score, stromal score, and immune score were calculated using ESTIMATE. (B–D) The distribution of tumor purity (B), stromal score (C), and immune score (D) in the three immune subtypes were compared, respectively. *** $P < 0.001$. ESTIMATE, Estimation of STromal and Immune cells in MAlignant Tumor tissues using Expression data.

immune checkpoint genes were all remarkably lower in subtype 1 and remarkably higher in subtype 3 ($P < 0.001$) (Figs. 2A–2H). This indicated that the immunophenotype of our hierarchical clusters was clearly distinguished, and that ovarian cancer subtype 3 may respond more effectively to checkpoint inhibitor therapy.

## Analysis of the TMB and BRCA mutations among the three subtypes of ovarian cancer

TMB is a predictor of tumor behavior and immunological response in a diverse range of cancers (*Goodman et al., 2017*). In general, tumors with a high TMB have elevated levels of neoantigens, which play an important role in immunotherapy activities (*Goodman et al., 2017*; *Schumacher & Schreiber, 2015*). We mined the somatic mutation profiles of 436 ovarian cancer patients from the SNP data in TCGA using Mutect, and calculated the TMB using the enumeration of mutation events per million bases. We then analyzed the TMB between the three subtypes of ovarian cancer and found that the three subtypes were not significantly correlated with TMB ($p = 0.726$) (Fig. 3A).

An increasing number of studies have documented that targeted therapies can stimulate the immune response of the host. The relationship between BRCA mutations and immunity is of great interest at present. We analyzed the connection linking BRCA1 and BRCA2 mutations in the three subtypes. The data from BRCA1 and BRCA2 mutations were mined
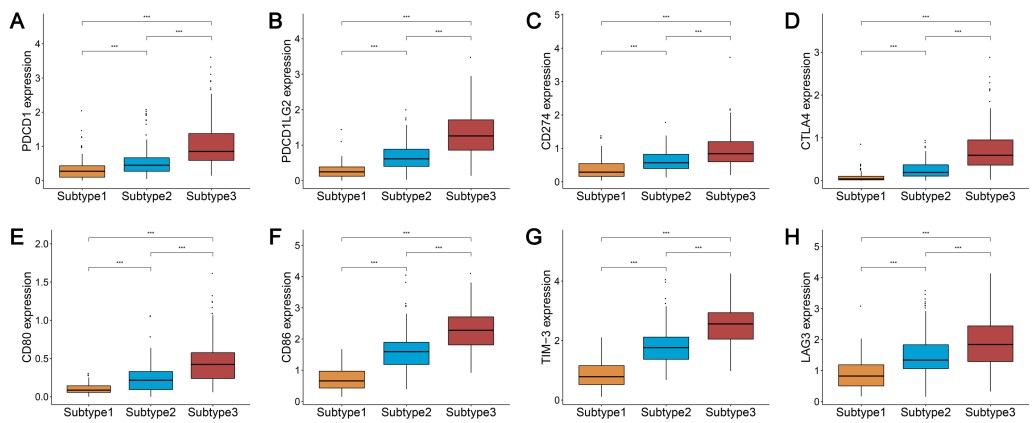

**Figure 2** Expression distribution of the eight immune checkpoint genes were all significantly lower and significantly higher in the subtype 1 and subtype 3, respectively. (A–H) PDCD1 (A), PDCD1LG2 (B), CD274 (C), CTLA4 (D), CD80 (E), CD86 (F), TIM-3 (G), LAG3 (H). *** $P < 0.001$.

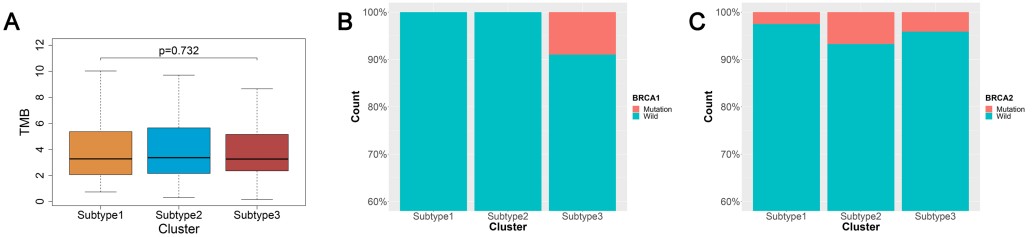

**Figure 3** TMB and BRCA mutation among the three subtypes of ovarian cancer. (A) Three subtypes were not significantly correlated with TMB. (B) All patients with BRCA1 mutations were concentrated in the subtype 3 and the difference was significant ($\chi^2$ test, $P = 0.0016$). (C) The patients with BRCA2 mutations were mainly found in the subtype 2 and subtype 3, but the difference was not significant ($\chi^2$ test, $P = 0.577$). TMB, tumor mutation burden.

from the SNP data in TCGA via Mutect. There were 23 patients with the BRCA1 mutation and 20 patients with the BRCA2 mutation out the 436 ovarian cancer patients sampled. We found 13 patients with the BRCA1 mutation and 13 patients with the BRCA2 mutation out of 274 ovarian cancer patients using the intersection between the mutation data and the immunity cluster data samples. Surprisingly, all BRCA1 mutation patients were in subtype 3, and the difference was significant ($\chi 2$ test, $P = 0.0016$) (Fig. 3B). The BRCA2 mutation ratio was greater in subtype 2 and subtype 3 compared with subtype 1, but this difference was not statistically significant ($\chi 2$ test, $P = 0.577$) (Fig. 3C).

## Different immune cells among the 3 subtypes of ovarian cancer

CIBERSORT can deduce 22 types of human immune cells, such as B cells, myeloid subset cells, T cells, NK cells, macrophages, and DCs, according to the gene expression data, using the gene-based deconvolution algorithm method (*Newman et al., 2015*). We set 1000 permutations and $P < 0.05$ as the maxim for the successful deconvolution of a sample and found that CD8 T cells, CD4 memory activated T cells, Tregs, macrophages M1, and resting

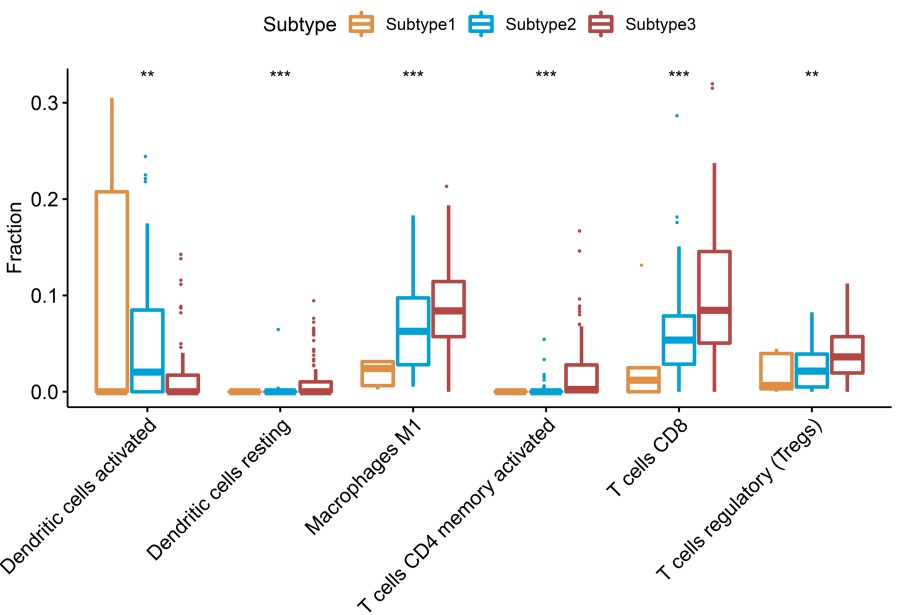

**Figure 4 Differential proportions of the immune cells in the three ovarian cancer subtypes.** Resting dendritic cells, macrophages M1, CD4 memory activated T cells, CD8 T cells, regulatory T cells were highest in the subtype 3 and lowest level in the subtype 2, but activated dendritic cells had an opposite trend. ** $P < 0.01$, *** $P < 0.001$.

dendritic cells were all at the highest levels in subtype 3 and the lowest levels in subtype 1 ($P < 0.01$). However, activated dendritic cells showed an opposite trend (Fig. 4).

## Prognostic analysis of the ovarian cancer subtypes and immune-associated gene sets

Survival analyses indicated significantly different prognoses among the three ovarian cancer subtypes. Subtype 2 had the worst survival prognosis among the three subtypes and there was no significant difference in survival between subtypes 1 and 3 (Fig. 5A). We analyzed the prognostic value of the different immune gene set expression scores for predicting patient survival. We found that the high expression level of the check-point, major histocompatibility complex (MHC) class I, APC co-inhibition, T cell co-inhibition, Th1 cells, Th2 cells, Tfh, inflammation-promoting, and Tregs was associated with a significantly better prognosis ($P < 0.05$) compared with low expression levels (Figs. 5B–5J).

## Identification of the ovarian cancer subtype-specific pathways and GO

GSEA identified 628 GO terms and 56 KEGG terms in subtypes 1 and 3. GO analysis indicated that the immunoglobulin complex, circulating immunoglobulin complex, MHC class II protein complex, immunoglobulin receptor binding, and the MHC protein complex were the top five significantly enriched biological processes in subtype 3. In addition, glucuronidation, metabolic processes, and methyl-CpG binding were the most enriched terms in subtype 1 (Figs. 6A and 6B). GSEA showed that the immune-associated pathways were most active in subtype 3, and consisted of Th17 cell differentiation, NF-κB signaling

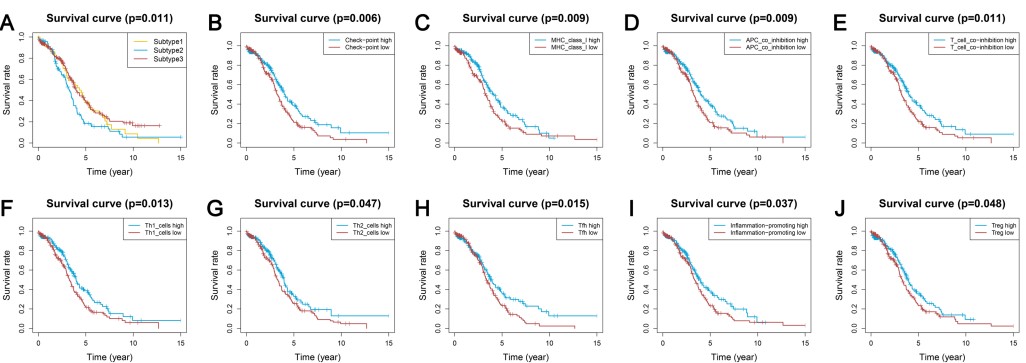

**Figure 5** **Kaplan-Meier curves showing survival prognosis of the ovarian cancer subtypes and immune-associated gene sets.** (A) The subtype 2 showed the worst survival prognosis among the three subtypes. (B–J) High level gene expression score of check-point (B), major histocompatibility complex class I (C), APC co-inhibition (D), T cell co-inhibition (E), Th1 cells (F), Th2_cells (G), Tfh (H), inflammation-promoting (I), Treg (J) were associated with a better prognosis. Treg, regulatory T cells.

axis, the B cell receptor signaling cascade, the T cell receptor signaling cascade, PD-L1 expression and the PD-1 checkpoint axis in cancer, the IL-17 signaling cascade, and the tumor necrosis factor (TNF) signaling axis. Our results verified that immune activity was elevated in subtype 3. However, subtype 1 was enriched in pathways, including maturity onset diabetes of the young, ascorbate and aldarate metabolism, pentose and glucuronate interconversions, fat digestion and absorption, and porphyrin and chlorophyll metabolism (Figs. 6C and 6D). These cascades may be inversely linked to ovarian cancer immunity.

### Validation of external datasets

We used the same method to hierarchically cluster ovarian cancer in the GSE51088 dataset, which included 172 ovarian cancer samples. Interestingly, there was a similar clustering result, with three distinct clusters (Fig. 7A). The immune and stromal scores were remarkably higher in subtype 3 and were markedly lower in subtype 1, while tumor purity showed an opposite result (Figs. 7B–7D). Most HLA genes and CD8A, CD1A, CD45R, and IL3RA expression levels were significantly lower in subtype 1 and significantly higher in subtype 3, which was consistent with the TCGA datasets (Figs. S2A and S2B). The expressions of the immune checkpoint genes, including PDCD1, CD274, CTLA4, CD80, CD86, TIM-3, and LAG-3 were all remarkably lower in subtype 1 and were significantly higher in subtype 3 (Figs. 7E–7L). These results were similar to results from previous research. Our results suggest that there are varied immune status subtypes in ovarian cancer that may affect the treatment of immune checkpoints differently.

### DISCUSSION

An increasing number of studies have identified ovarian cancer subtypes based on genomic profiling to provide individualized treatments and improve patient survival (*Schwede et al., 2020*; *Yang et al., 2018*; *Zheng et al., 2020*). However, few studies have classified ovarian cancer based on immune signatures. We sought to identify immune-correlated ovarian

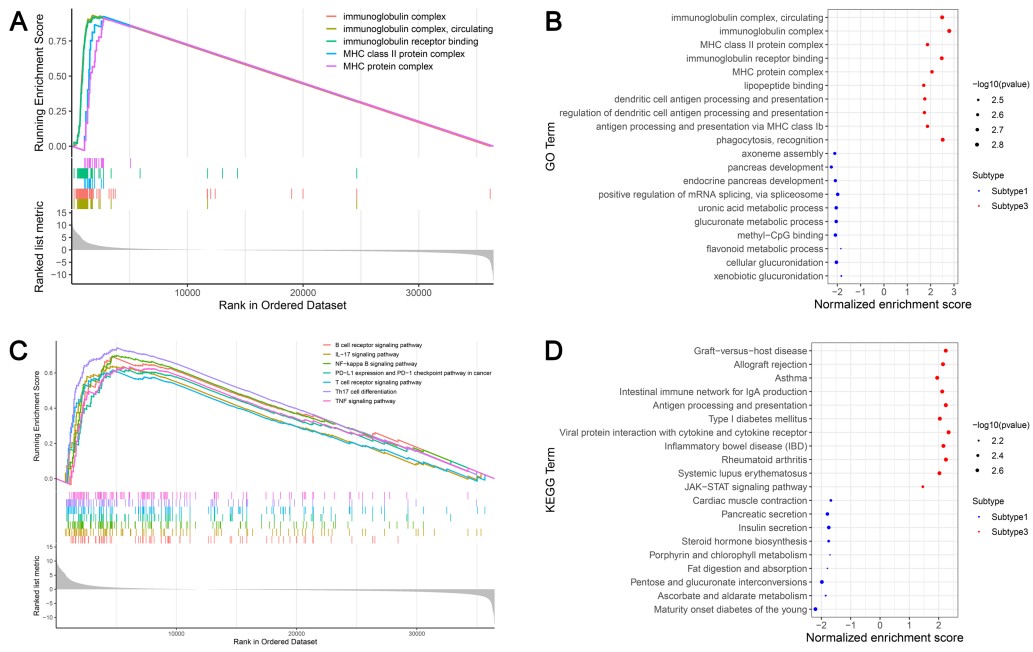

**Figure 6** **GSEA identified GO and KEGG pathways enriched in the subtype 1 and subtype 3.** (A) GO analysis of the top five significantly enriched biological processes in the subtype 3. (B) GO analysis of the top 10 biological processes significantly enriched in the subtype 1 and the subtype 3, respectively. (C) KEGG analysis of the subtype-specific pathways enriched in the subtype 3. (D) KEGG analysis of the top 10 pathways significantly enriched in the subtype 1 and the subtype 3, respectively. GO, Gene Ontology; KEGG, Kyoto Encyclopedia of Genes and Genomes.

cancer subtypes in a TCGA-ovarian cancer cohort based on 29 immune-associated gene sets, which typified different immune cell types, functions, and pathways. We classified ovarian cancer into three subtypes using ssGSEA, with an immune score ranging from low to high. Our results were reproduced in the external dataset GSE51088.

The immune microenvironment of subtype 3 was strengthened, with greater levels of immune cell invasion and stronger anti-tumor immune activities, such as high levels of cytotoxic T cells and B cell invasion. The expression levels of most of the HLA genes were highest in subtype 3. The recognition and expression of tumor antigens on effector cells, such as CD8 + T cells, was the threshold of the immune response. HLA played a central role in providing effector CD8 + T cells with natural intracellular proteins or neoantigens produced by the cancer cells (*Koşaloğlu-Yalçın et al., 2018*). The down regulation of HLA class I expression participated in the departure from the host immune system and immunotherapy resistance (*Chowell et al., 2018*; *Lhotakova et al., 2019*). Many immune cell subgroup maker genes, including FOXP3 (Treg), CD45RO (memory T cell), CD8A (cytotoxic T cell), CD20 (B cell), CD1A (iDC), CXCR5 (Tfh cell), IL3RA (pDC) were stronger in subtype 3. We found that CD8 T cells, CD4 memory activated T cells, Tregs, macrophages M1, and resting dendritic cells were higher in subtype 3 and lower in subtype 1. Our results further confirmed that there were different immune subtypes in ovarian cancer and that the immune activity of subtype 3 was stronger. Survival analyses showed

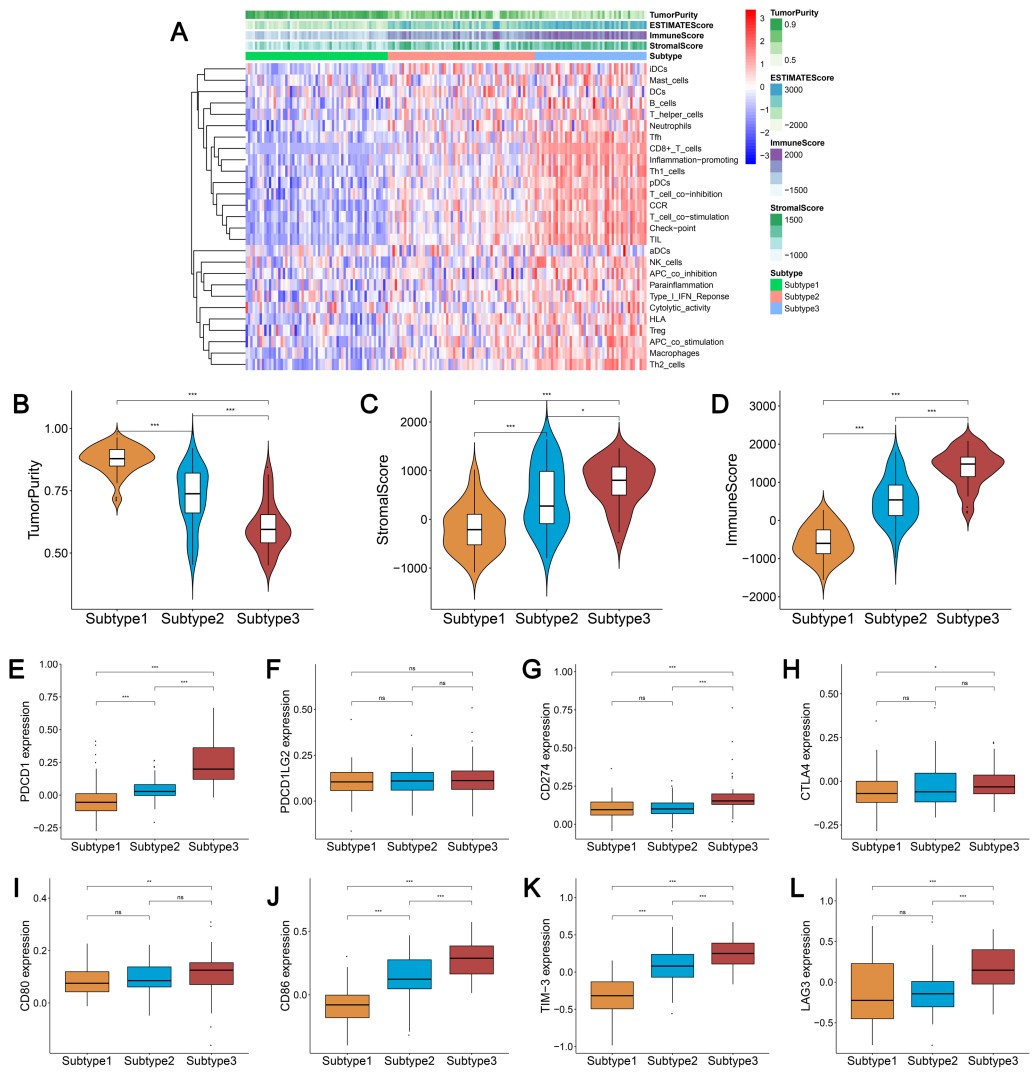

**Figure 7** **Validation of the external datasets.** (A) Hierarchical clustering of ovarian cancer yields three subtypes in the GEO dataset. Red represents high expression and blue represents low expression. (B–D) The distribution of tumor purity (B), stromal score (C), and immune score (D) were compared in the three immune subtypes in the GEO dataset, respectively. (E–L) Expression distribution of the eight immune checkpoint genes in the three ovarian cancer subtypes in the GEO dataset. PDCD1 (E), PDCD1LG2 (F), CD274 (G), CTLA4 (H), CD80 (I), CD86 (J), TIM-3 (K), LAG3 (L) * $P < 0.05$, ** $P < 0.01$, *** $P < 0.001$. GEO, Gene Expression Omnibus.

that the worst prognoses were found in subtype 2 and there was no significant survival difference between subtypes 1 and 3. This suggested that immune-enhanced subtypes may not have the best outcome in ovarian cancer, and was consistent with the findings from *Zheng et al. (2020)*.

A number of recent studies have demonstrated that the immune checkpoint has a pivotal role in the immunosuppression of cancer. It is well-known that PD-1, CTLA4, LAG-3, TIM-3, BTLA, and VISTA are the most common immune checkpoint receptors.

It was previously reported that blocking PD1/PD-L1 in combination with other agents, particularly other checkpoint suppressors was more effective for immunosuppression (*Boutros et al., 2016*; *Doo, Norian & Arend, 2019*; *Huang et al., 2017*). Clinical studies have shown that the effect of treatment in patients with advanced melanoma could be improved when combined with the anti-PD-1/PD-L1 antibody and the CTLA-4 inhibitor (*Boutros et al., 2016*). Blocking PD-1 alone was shown to be insufficient in controlling murine ovarian tumor growth but dual blocking of the PD-1-LAG-3 or PD-1-CTLA-4 cascades could delay murine ovarian tumor growth, and blocking 3 PD-1-CTLA-4-LAG-3 cascades was superior if the PD-1 pathway was entirely blocked (*Huang et al., 2017*). We identified that the expression levels of the checkpoint genes, including PDCD1, CD274, PDCD1LG2, CTLA4, CD86, CD80, LAG-3 and TIM-3 were remarkably higher in subtype 3, indicating that subtype 3 may be linked to the intrinsic immune response of ovarian cancer. This may provide new insights for the treatment of ovarian cancer with immune checkpoint blockers.

Many studies have found that a higher level of TMB was associated with higher neoantigen loads, which are the target of ICIs (*Brown et al., 2014*; *Gubin et al., 2014*; *Samstein et al., 2019*). TMB generates new antigens resulting in the enrichment of the immune cells in tumors and could predict survival across diverse kinds of human cancer, (e.g., non-small-cell lung cancer, melanoma, and bladder cancer). These results are applicable in patients being treated by anti-CTLA-4 or anti-PD-1 therapies (*Samstein et al., 2019*). We failed to detect an association between TMB and tumor-infiltrating immune cells and found no significant difference in TMB among the three immune ovarian cancer subtypes. Similarly, *Dai et al. (2018)* found no association between TMB and the tumor immune response, represented by cytolytic activity or immune cell infiltration. Therefore, TMB may not be an appropriate biomarker for ovarian cancer immunotherapy. We found that all patients with BRCA1 mutations were in subtype 3 and patients with BRCA2 mutations were primarily in subtypes 2 and 3. Ovarian cancer with BRCA1 or BRCA2 mutations had increased immune infiltrates compared to those without mutations (*McAlpine et al., 2012*). *Strickland et al. (2016)* demonstrated that BRCA 1/2-mutated high grade serous ovarian cancer had remarkably elevated CD3+ and CD8+tumor-infiltrating lymphocytes and elevated expression levels of PD-1. PD-L1 in the tumor-linked immune cells contrasted with that in homologous recombination proficient tumors. Another study also showed that the presence of intraepithelial CD8+ T-cells was linked to a mutation or loss of expression of BRCA1 (*Clarke et al., 2009*). These findings suggest that BRCA-mutated ovarian cancer may be more sensitive to immune checkpoint blockade therapy.

We identified 628 GO and 56 KEGG terms in subtypes 1 and 3 using enrichment analysis. GO analysis indicated that the immunoglobulin complex, the MHC class II protein complex, immunoglobulin receptor binding, and the MHC protein complex were primarily enriched in subtype 3. T cell immunity requires the recognition of antigens in the context of MHC class I and class II proteins by CD8+ and CD4+ T cells, respectively (*Koşaloğlu-Yalçın et al., 2018*). A previous study found that the MHC proteins confer differential sensitivity to CTLA-4 and PD-1 blocking in melanoma (*Rodig et al., 2018*). The immune-linked cascades were most active in subtype 3, and included Th17 cell

differentiation, the NF-$\kappa$B signaling cascade, the B cell receptor signaling axis, the T cell receptor signaling axis, PD-L1 expression, the PD-1 checkpoint cascade, the IL-17 signaling axis, and the TNF signaling pathway. The NF-$\kappa$B signaling axis has been proven to be the major cascade involved in ovarian cancer, enhancing chemoresistance, cancer stem cell maintenance, metastasis, and immune evasion (*Harrington & Annunziata, 2019*). *Bilska et al. (2020)* determined the proinflammatory nature of the ovarian cancer microenvironment with high levels of IL-17A in the peritoneal fluid and a high percentage of Th17 infiltration and suggested that Th17 cells/IL-17A may serve an advantageous role in ovarian cancer immunity. A number of studies have proved that the PD-1/PD-L1 cascade, B cell and T cell receptor signaling axes, and TNF signaling cascades were associated with ovarian cancer immunity (*Ghisoni et al., 2019*; *Gupta et al., 2019*; *Josephs et al., 2017*).

There were some limitations to our study. Firstly, our data was from public repositories and was not self-generated. Secondly, the BRCA1 and BRCA2 mutation ratios in ovarian cancer were relatively low in the SNP data from TCGA. Thus, further research using a larger sample size should be conducted to validate the relevance of BRCA mutations with ovarian cancer immunity. Finally, immunogenomic analysis requires more experimental evidence to verify the role of BRCA1/BRCA2 mutations, checkpoint genes, and the enriched cascades involved in the immune microenvironment.

We identified ovarian cancer subtypes based on immune signatures which were distinct in the tumor microenvironment, immune cells, immune checkpoint molecules, BRCA mutations, and clinical prognoses. These findings may help develop novel immunotherapy strategies in ovarian cancer.

### Funding

This work was supported by the National Natural Science Foundation of China (grant no. 81760466). The funders had no role in study design, data collection and analysis, decision to publish, or preparation of the manuscript.

### Grant Disclosures

The following grant information was disclosed by the authors:
National Natural Science Foundation of China: 81760466.

### Competing Interests

The authors declare there are no competing interests.

### Author Contributions

- Yousheng Wei conceived and designed the experiments, performed the experiments, analyzed the data, prepared figures and/or tables, authored or reviewed drafts of the paper, and approved the final draft.
- Tingyu Ou performed the experiments, analyzed the data, prepared figures and/or tables, authored or reviewed drafts of the paper, and approved the final draft.

- Yan Lu, Guangteng Wu, Ying Long and Xinbin Pan performed the experiments, prepared figures and/or tables, and approved the final draft.
- Desheng Yao conceived and designed the experiments, authored or reviewed drafts of the paper, and approved the final draft.

## Data Availability

The raw data and code are available in the Supplementary Files.

Gene expression profiles and clinical data are available at TCGA. Search terms: "Primary Site" IS "ovary" AND "Program" IS "TCGA" AND "Project" IS "TCGA-OV" AND "Data Category" IS "Transcriptome Profiling" AND "Data Type" IS "Gene Expression Quantification" AND "Workflow Type" IS "HTSeq-FPKM".

The somatic mutation data is also available at TCGA. Search terms: "Primary Site" IS "ovary" AND "Program" IS "TCGA" AND "Project" IS "TCGA-OV" AND "Data Category" IS "Simple Nucleotide Variation" AND "Data Type" IS "Masked Somatic Mutation" AND "Workflow Type" IS "Mu Tect2 Variant aggregation and Masking".

The expression data of the validation dataset is available at NCBI GEO: GSE51088.

## Supplemental Information

Supplemental information for this article can be found online at http://dx.doi.org/10.7717/peerj.10414#supplemental-information.

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
