# Peer review of "Classification of ovarian cancer associated with BRCA1 mutations, immune checkpoints, and tumor microenvironment based on immunogenomic profiling"

_PeerJ, doi:10.7717/peerj.10414_

## Round 0.1 · original submission · Major Revisions

All critiques of both reviewers should be addressed and the manuscript should be addressed accordingly.

·

Basic reporting

The author provides background that the therapeutic effect in ovarian cancer is not satisfactory. References is also sufficient. However, there are some gramma errors.
In the PDF file:
1. Methods part on page 4. ‘we classified ovarian cancer…’, ‘w’ in ‘we’ needs to be capitalized.
2. Results part on page 4. ‘Interestingly, though tumor mutation burden (TMB) was no significant
difference among three subtypes, but all patients with BRCA1 mutations were detected in Immunity_H
subtype.’ This sentence doesn’t read well, it is wrong in gramma. Should be ‘was not significantly different’. Also, usually ‘though’ and ‘but’ is not used together.
3. Line 45 United Stated->United States
4. Line 48 ‘Although we have made…’, it’s better to say ‘Although researchers have made…’ if the cited reference is not the author’s work.
5. Line 67-68, ‘In the phase II…. show that…’, this sentence has error in gramma.
6. Line 77-78, ‘ssGSEA was used…, then classified…’, this sentence has error in gramma.
7. Line 81-82, ‘Our study findings …immunotherapy’, this sentence has error in gramma.
8. Line 213 ‘the same mothed’ should be ‘the same method’.

This article also has some other gramma errors in Result section starting from line 136, the authors need to carefully double check before re-submission.

Experimental design

The experiment design was good. The research question is well defined, the authors wanted to check the immune landscape of patients to improve treatment efficacy. The research could also fill the gap in the treatment.
The methods are also described in details.

Validity of the findings

The authors provided the validation of results with another dataset and shows consistency. The results were clearly explained. Figures and legends are also clear. But it would be great if the authors can provide more description of the figures. For example, figure 2 and 5.
In the result for BRCA1 mutation and BRCA2 mutation, the subgroup number for BRCA2 mutation is 11 out of ~430 total patients. And then 11 patients distribute in 3 immunity groups. Even the statistical test result shows that the difference is not significant, 11 is a small group number. It would be great if the author could find larger datasets to confirm this.

Reviewer 2 ·

Basic reporting

.

Experimental design

.

Validity of the findings

.

Additional comments

1. Please define the following terms in the text: immune score, tumor purity, stromal score, estimate score. How are they derived in the ESTIMATE package? What do they mean? What is the importance of the numbers?
2. Line 45: Missing a 'the' before the United States.
3. Line 45: Due to a lack of symptoms...
4.  Line 62: Give the full names of CTLA-4, PD-1, and PD-L1.
5. Line 62: ...PD-L1 receptors have...
6. Line 63: ...malignant melanoma, lung cancer, and bladder cancer...
7. Line 65: However, the response rate to ICIs for patients with ovarian cancers remains unsatisfying, in which the objective response rate (ORR) was less than 15%.
8. Line 77: It should be cancer and not the cancer. 
9. Line 139-141: Please restructure this sentence. 
10. Line 142: ...we got the immune score, stromal score, ESTIMATE scores, and tumor...
11. Line 145: These results indicated that immune and stromal cellnumbers...
12. Line 154: ...CTLA4, TIM-3, and LAG-3...
13. Line 163: ...important role...
14. Line 166: ...3 subtypes ovarian cancer were...
15. Line 169: ...have shown...
16. Line 194-196: How are the parameters mentioned associated with a better prognosis? This statement is not clear.  
17. LIne 210: what is the function of chlorophyll metabolism in human cells? How can a human cell line have an enrichment of the chlorophyll-metabolism pathway?  
18. Line 214: ...a similar...
19. Line 246: ...immune checkpoint plays a crucial role... 
20. Line 251: Restructure the sentence.
21. Line 267: What is tumor immune? Tumor immune cells?
22. Line 269: Sentence not clear. 
23. Please use either tumour or tumor. There are both spellings present in this manuscript. Please check with the journal recommendations for British vs. American spellings. 
24. Figure 1. Define red-->blue gradient. How is the tree constructed? Figure legends should be descriptive enough to sufficiently understand the figure.   
25. Figures 2 and 3. Please provide a descriptive figure legend. Also, the texts in the graphs are too small to read properly. 
26. Figure 4. Once again, a proper figure legend is missing. The authors should explain the terms of activated and resting dendritic cells. What type of plot is this? How about error bars? How was the statistical significance calculated?
27. Figures 5 and 6. Texts are too small. 
28. Figure 6: The term enrichment score should be defined mathematically. Does this plot only convey a list of pathways? Anything more?  

The authors used ssGSEA projections to enrich a number of genes associated with ovarian cancer from the TCGA database. They divided the ovarian cancers into 3 classes based on their respective ssGSEA scores and analyzed these classes at finer details. In my view, methodology, experimentation, and results of this study are valid. What needs work is the writing and language. Also, the authors wrote this manuscript like a list of genes/pathways that are enriched in the different sub-types and definitely missed the opportunity to functionally analyze the data. If the sub-types are formed based on immune scores, then isn't it expected that they would have differential enrichment of immune genes? The title of the paper emphasized on the BRCA1 mutations, but that mutant pool is only >3% of the ovarian cancer dataset. The authors need to explain these issues better in the discussion.

---

## Round 0.2 · Minor Revisions

Please address the remaining critiques of the reviewers. Please note that reviewers indicated that your manuscript requires some editorial work to fix linguistic issues.

·

Basic reporting

The authors addressed the comments.

New comments:
In results part, 'Analysis of the TMB and BRCA mutations among the 3 subtypes of ovarian cancer'. The authors mentioned Fig. 3A, FIg 3B and 3C. However, in Fig. 3, there is no A, B, C parts. I would suggest the authors to double check the figures.

Experimental design

The authors addressed the comments. The paper is fine for this part.

Validity of the findings

The comments are addressed. The paper is fnie for this part.

Reviewer 2 ·

Basic reporting

.

Experimental design

.

Validity of the findings

.

Additional comments

I am happy with the revisions and the rebuttals. I would still suggest the authors revise their manuscripts for typos and grammatical errors or take editorial help. I am happy to recommend this revised manuscript for publication.

---

## Round 0.3 · accepted · Accept

Thank you for addressing the remaining critiques of the reviewers.